# Structural basis for the intracellular regulation of ferritin degradation

Fabian Hoelzgen[1,2,9], Thuy T. P. Nguyen[3,9], Elina Klukin[2], Mohamed Boumaiza[4], Ayush K. Srivastava[4], Elizabeth Y. Kim[3], Ran Zalk [5], Anat Shahar[5], Sagit Cohen-Schwartz[6], Esther G. Meyron-Holtz[7], Fadi Bou-Abdallah [4], Joseph D. Mancias [3,8] ✉ & Gabriel A. Frank [2,5,6] ✉

The interaction between nuclear receptor coactivator 4 (NCOA4) and the iron storage protein ferritin is a crucial component of cellular iron homeostasis. The binding of NCOA4 to the FTH1 subunits of ferritin initiates ferritinophagy —a ferritin-specific autophagic pathway leading to the release of the iron stored inside ferritin. The dysregulation of NCOA4 is associated with several diseases, including neurodegenerative disorders and cancer, highlighting the NCOA4-ferritin interface as a prime target for drug development. Here, we present the cryo-EM structure of the NCOA4-FTH1 interface, resolving 16 amino acids of NCOA4 that are crucial for the interaction. The characterization of mutants, designed to modulate the NCOA4–FTH1 interaction, is used to validate the significance of the different features of the binding site. Our results explain the role of the large solvent-exposed hydrophobic patch found on the surface of FTH1 and pave the way for the rational development of ferritinophagy modulators.

Iron ions serve as cofactors in many biochemical processes, where their versatile redox chemistry plays a central role in the enzymatic activity of processes such as DNA replication and energy production[1]. However, this redox activity can also lead to the formation of reactive oxygen species via Fenton-like reactions resulting in cell injury and death by oxidative damage thereby demanding tight regulation of cellular iron levels[2]. The regulation of cellular iron levels is achieved by iron regulatory proteins (IREBP2, ACO1) that sense intracellular labile iron levels and exert control over the import, export, and storage of iron, thus securing continuous iron supply while preventing cytotoxicity associated with uncontrolled cellular iron levels[3]. A central component of intracellular iron homeostasis is the iron storage protein ferritin that forms a 24-mer cage-like complex composed of two structurally similar, but functionally different subunits, FTL and FTH1. Each ferritin oligomer can sequester up to ~4000 ferric iron ions inside its cavity as an inorganic mineral core[4]. Iron sequestered in ferritin is reintroduced into the cytoplasm by 'ferritinophagy', a selective autophagy pathway that involves the trafficking of iron-loaded ferritin to the lysosome where its content is released[5–7]. The initial stage of ferritinophagy involves the binding of Nuclear Receptor Coactivator 4 (NCOA4) to FTH1 subunits in the ferritin oligomer[7,8]. This process leads to the agglomeration of a ferritin-NCOA4 condensate in the cytoplasm that is subject to macroautophagy for lysosomal degradation or storage as hemosiderin and endosomal microautophagy for extracellular release[9].

[1]The Kreitman School of Advanced Graduate Studies, Marcus Family Campus, Ben-Gurion University of the Negev, Beer-Sheva, Israel. [2]Department of Life Sciences, Marcus Family Campus, Ben-Gurion University of the Negev, Beer-Sheva, Israel. [3]Division of Radiation and Genome Stability, Department of Radiation Oncology, Dana-Farber Cancer Institute, Harvard Medical School, Boston, MA, USA. [4]Department of Chemistry, State University of New York at Potsdam (SUNY Potsdam), Potsdam, NY, USA. [5]Ilse Katz Institute for Nanoscale Science & Technology, Marcus Family Campus, Ben-Gurion University of the Negev, Beer-Sheva, Israel. [6]The National Institute for Biotechnology in the Negev – NIBN, Marcus Family Campus, Ben-Gurion University of the Negev, Beer-Sheva, Israel. [7]Faculty of Biotechnology and Food Engineering, Technion, Haifa, Israel. [8]Department of Radiation Oncology, Brigham and Women's Hospital, Harvard Medical School, Boston, MA, USA. [9]These authors contributed equally: Fabian Hoelzgen, Thuy T. P. Nguyen. ✉e-mail: Joseph_Mancias@dfci.harvard.edu; frankg@bgu.ac.il

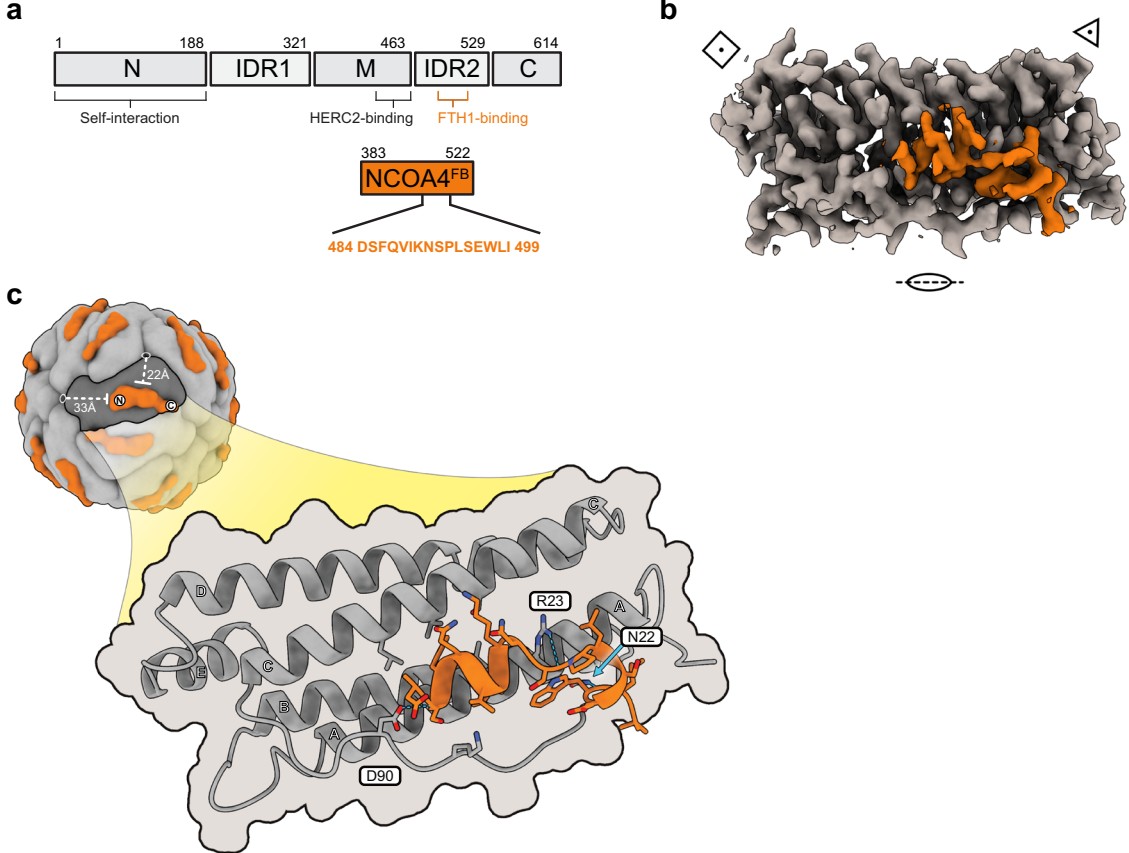

**Fig. 1 | The overall organization of the NCOA4FB-FTH1 complex. a** Schematic representation of the domain organization of NCOA4. The NCOA4FB segment is below in an orange box, and the 16 amino acids mediating the interaction with FTH1 are designated with a one-letter code. N designates the self-interacting N-terminal domain, The two intrinsically disordered regions are designated by IDR1 and IDR2 respectively, M- designates the middle domain, and C- the C-terminal domain (based on Ohshima et al.[9]). **b** The asymmetric unit of the EM-density map of the NCOA4FB-FTH1 complex (EMD-18658). NCOA4FB is orange. The C2, C3, and C4 symmetry axes are designated by an oval, triangle and square respectively.

**c** Schematic representation of the position of NCOA4FB with respect to the geometry of the nanocage and the detailed molecular model of the interaction. The structurally resolved segment of NCOA4FB is positioned in an elongated cleft running across the long axis of each FTH1 monomer formed by helix C and the B-C loop (the helices of FTH1 are designated by capital letters). NCOA4FB is relatively far from the 3- and 4-fold symmetry axes of the nanocage (22 and 33 Å respectively) and is localized to a single subunit of FTH1. Three H-bonds between FTH1 and NCOA4FB stabilize their interaction (N22-W497, R23-S492, and D90-S485).

NCOA4 is mostly intrinsically disordered, and the structural basis for its interaction with ferritin is unknown. Furthermore, despite the similarity between FTH1 and FTL, NCOA4 binds specifically to FTH1 but not FTL. An alanine mutational scan of surface residues conserved among FTH1 orthologs but not FTL revealed that the FTH1-R23 is crucial for the NCOA4-FTH1 interaction. Truncation mutants of NCOA4 identified NCOA4-383-522 as the FTH1 interacting portion of NCOA4 and alanine scanning mutational analysis of the NCOA4-488-499 fragment identified I489 and W497 as important residues in mediating the NCOA4-FTH1 interaction[10]. However, other than this R23 single amino acid on FTH1 and several amino acids on NCOA4, factors determining the specificity between the NCOA4 ferritin-binding fragment (amino acids 383-522, here on NCOA4FB) and FTH1 remained unknown.

Here, we show the cryo-electron microscopy (Cryo-EM) structure of the NCOA4FB in complex with FTH1. The structure explains the strong binding between the two proteins, the pivotal role of FTH1-R23 and NCOA4-I489 and W497 in this interaction, and the origin of the selectivity of NCOA4 towards FTH1 over FTL. To corroborate our structural analysis, we design several FTH1 mutants that based on the structure we predict to modulate the interaction between FTH1 and NCOA4 as well as a rationally designed FTL mutant that we predict to bind NCOA4. We interrogate the effects of these mutants using biophysical assays, and cellular interaction and localization assays,

thereby confirming the structure-function of the NCOA4-FTH1 binding interaction. This study defines the structural underpinnings of NCOA4-mediated ferritinophagy and will facilitate future work to understand the biochemical regulation of the NCOA4-FTH1 interaction as well as efforts to modulate the interaction for potential therapeutic applications.

## Results

### The structure of the NCOA4-FTH1 complex

The 3D-EM map of the NCOA4FB-FTH1 complex was determined by Cryo-EM (Fig. 1a, b and Supplementary Fig. 1, EMD-18658). The overall density of the NCOA4FB segments in the 3D-EM map was lower than the density of the FTH1 region. This can be explained by NCOA4FB:FTH1-subunits occupancy ratio of ~1:3 found in previous studies[11]. As expected, due to the intrinsically disordered nature of the ferritin binding region of NCOA4[9,12] (Fig. 1a), the map revealed only the structure of the 16 amino acids of NCOA4 (aa484-499) that were tightly bound to the FTH1 surface (Fig. 1, and Supplementary Fig. 1-2, Supplementary Table 1 and PDB 8QU9). NCOA4 interacts with FTH1 through two helical segments connected by P493 (Figs. 1c and 2). The first segment (S485-S492) consists of a two-turn α-helical structure that is capped with a H-bond from S492 (Fig. 2a, b), and the second segment (P493-L498) consists of ~2 turns of a distorted 3₁₀-helix (Fig. 2a, c). NCOA4 is anchored to a cleft in FTH1, which harbors these

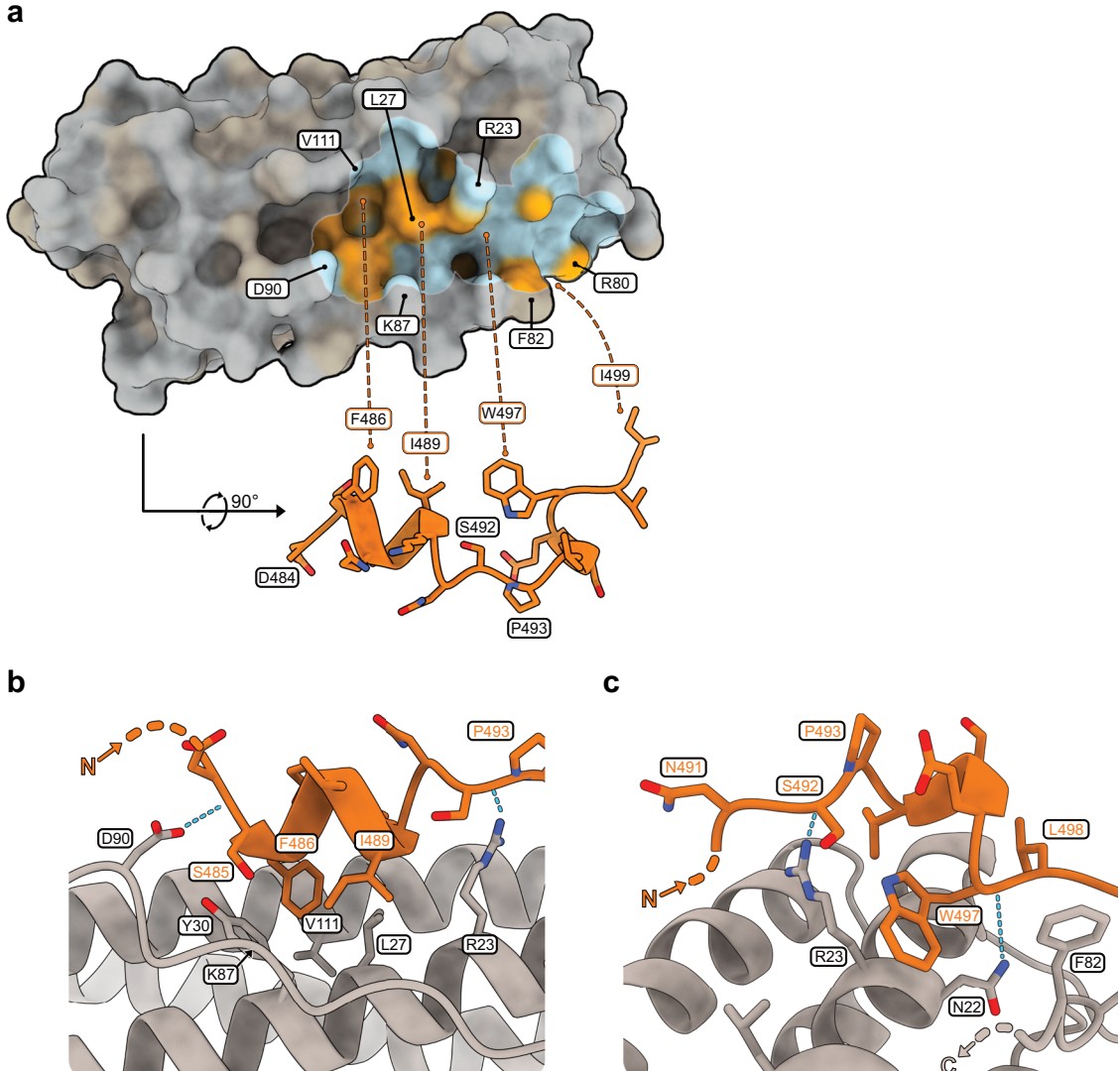

**Fig. 2 | The molecular interface of the NCOA4^FB-FTH1 complex. a** The panel depicts the molecular surface of the FTH1 subunit in shades of gray. The footprint of NCOA4 on the molecular surface of FTH1 is highlighted by a white outline. The region inside NCOA4's footprint is colored according to the hydrophobicity of the FTH1 surface, from light blue to orange, hydrophilic to hydrophobic respectively[31,32]. The visible segment of NCOA4 is orange. The segment was shifted down and rotated -90° to reveal the hydrophobic amino acids of NCOA4 that interact with FTH1. The locations of these amino acids relative to the FTH1 surface are designated by dashed lines. The NCOA4^FB-FTH1 interface is stabilized by F82 and the hydrophobic segment of R80 holding I499 and three neighboring hydrophobic pockets holding F486, I489, and W497. **b** A focused view of the

hydrophobic pockets harboring the N-terminal segment of NCOA4^FB. The pockets are delineated from one side by Y30 and on the other side by the hydrophobic segment of the R23 residue. V111, L27 and the hydrophobic segment of K87 line the pockets from the middle (the position of the Cα of K87 is marked with an arrow, its residue is not shown to prevent it from blocking the field of view). **c** A focused view of the hydrophobic pockets harboring the C-terminal segment of NCOA4^FB. W497 is sandwiched between the hydrophobic segment of the R23 residue and F82 (The C-terminal side of the BC-loop after F82 is not shown to avoid blocking the view). Inside this hydrophobic region N22 interacts forms a hydrogen bond with the carboxyl oxygen of W497.

two helical segments. The cleft is confined from its sides by FTH1 Helix C and the long B-C loop; and from its bottom by Helix A (marked with capital letters in Fig. 1c)[4,13]. The central part of the cleft is lined with hydrophobic residues, while its edges are marked with two hydrophilic amino acids, N22 and D90, which form H-bonds with NCOA4 (Fig. 2).

On FTH1, the bulky R23 residue in Helix A bisects the cleft into two separate regions; one holds the α-helical segment of NCOA4, and the other holds NCOA4's 3_{10}-helical segment. P493 bridges above R23, thus connecting the two segments of NCOA4 while avoiding a clash with R23 (Fig. 2a, c). The guanidine group of R23 forms a hydrogen bond with the carbonyl oxygen of S492, likely contributing to the stability of the NCOA4-FTH1 interface. However, as this H-bond is solvent-exposed, this interaction alone cannot

explain the crucial role of R23 in stabilizing the NCOA4·FTH1 complex[10].

In addition to this H-bond, the aliphatic region of R23 takes part in two out of three back-to-back hydrophobic pockets that interdigitate with three hydrophobic amino acids of NCOA4 (W497, I489, and F486) (Fig. 2). One side of the aliphatic region of R23 interacts with W497, which is pinned into the first pocket, while the other side of R23 forms a pocket with the aliphatic region of K87 and L27, holding NCOA4's I489. The other side of L27 takes part in the third pocket (together with V111) that holds F486. D90, placed at the end of this multi-layered hydrophobic structure, is engaged in H-bonds with S485. These H-bonds mark the edge of the NCOA4-FTH1 interaction interface.

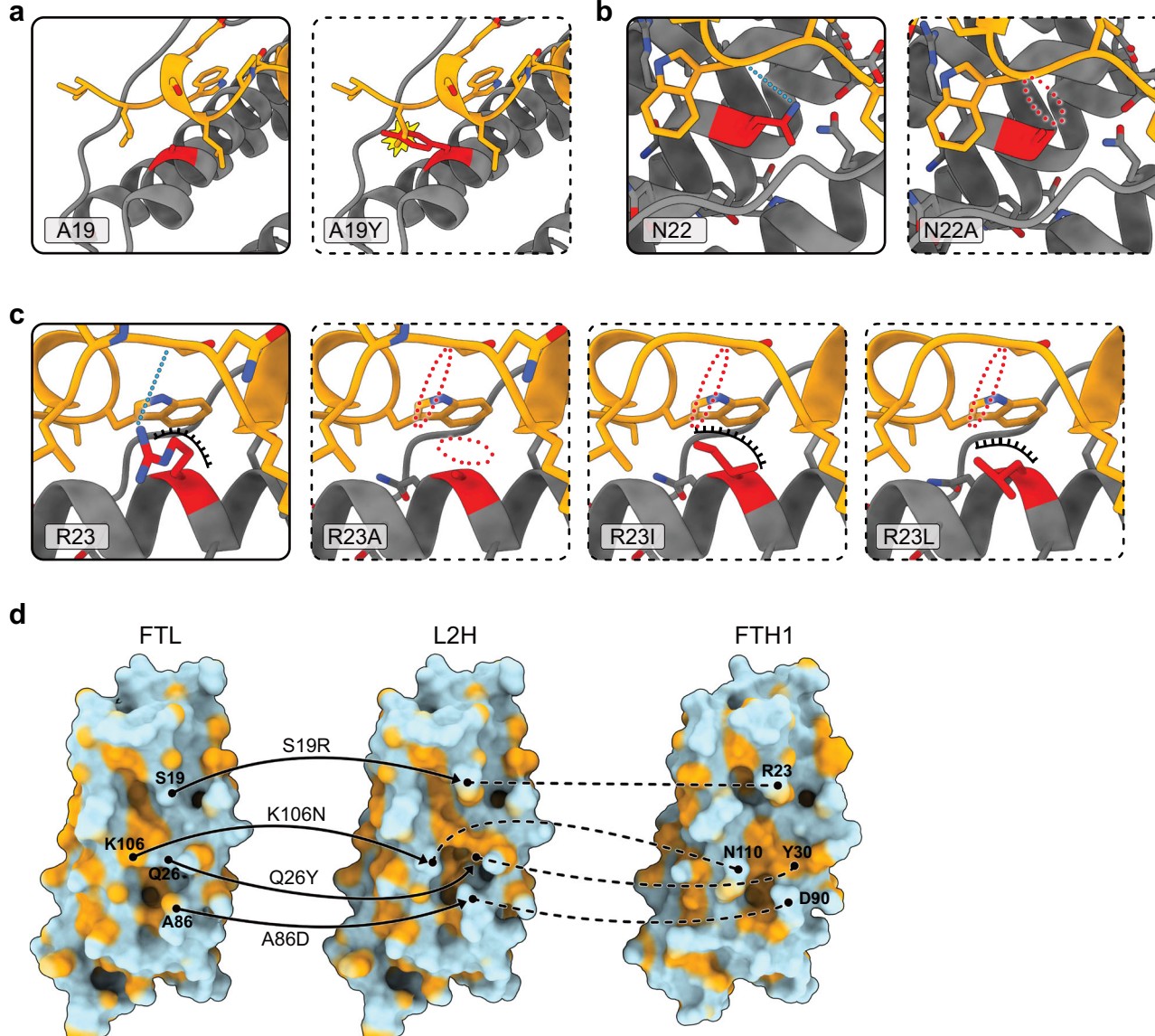

**Fig. 3 | Visualization of the mutations designed for modulating the interaction of FTH1 with NCOA4 and the structure-guided design of the binding site for NCOA4 on FTL. a–c** NCOA4 is orange, FTH1 is gray, and mutated amino acids are in red, dashed frames designate the presumed position of the amino acids after mutations. **a** A19Y: abolishes the NCOA4-FTH1 interaction by blocking the opening of the hydrophobic cleft that harbors NCOA4. **b** N22A: Removes one of the three H-bonds stabilizing the NCOA4-FTH1 interface **c** R23A: known to abolish the interaction[10]. R23I and R23L: preserves part of the hydrophobic pocket found in the *WT* but abolishes the H-bond. **d** the hydrophobic surface of *WT*-FTL on the left, *WT*-FTH1 on the right, and the designed L2H mutant in the middle. The mutations are designated on each panel. Visualization of the phosphomimetic mutations S114D and S114E and a visualization of the designed changes to the electrostatic surface of the gain of function mutant L2H are depicted in Supplementary Figs. 3 and 4, respectively.

## Structure-based design of mutations modulating the NCOA4-ferritin interactions

A library of FTH1 mutants with modulated NCOA4 binding properties can help understand how NCOA4 dictates the fate of FTH1 and the onset of ferritinophagy. To generate such a library and corroborate our structural analysis, we designed several FTH1 mutants aimed at rationally changing the interaction between NCOA4 and FTH1 (Fig. 3 and Table 1, and Supplementary Figs. 3–4). We analyzed the effect of these mutations on NCOA4-ferritin binding thermodynamics (Fig. 4, Supplementary Fig. 5) and by immunoprecipitation and colocalization in cells (Fig. 5, Supplementary Fig. 6).

Our structural findings highlight the hydrophobic nature of the aliphatic part of the R23 moiety as a major contributor to the stability of the NCOA4-FTH1 complex. To test this, we mutated FTH1-R23 to the hydrophobic amino acids leucine or isoleucine (R23L and R23I),

expecting that these mutations will weaken but not abolish the NCOA4-FTH1 interaction, in contrast to R23A that abolished the interaction[10] (Fig. 3c). To demonstrate the importance of the cleft structure of FTH1, we generated mutant A19Y which replaces a small amino acid at the edge of the cleft with the bulky tyrosine residue, thus blocking its entrance and reducing binding (Fig. 3a). The H-bond between N22 and the carbonyl oxygen of W497 is protected from the solvent. To test the importance of this interaction, we introduced the mutation N22A (Fig. 3b). Based on our structural analysis, S114 on the C helix of FTH1 is positioned at the NCOA4-FTH1 interface. While S114 does not appear to contribute to the interaction between the proteins, S114 phosphorylation has been observed in high throughput mass spectrometry-based phosphoproteome studies[14]. However, the effect of S114 phosphorylation on the NCOA4-FTH1 interaction is unknown. To test this possibility, we introduced the phosphomimetic mutations

**Table 1 | Comparison of the affinity measurements between NCOA4 and FTH1-WT and its mutants obtained by ITC, immunoprecipitation and colocalization assays**

| Mutant | Thermodynamic parameters from ITC | | | | | Relative interaction by immunoprecipitation | | Cells with colocalized mRuby3-FTH1/FTL:mGFP-NCOA4 puncta | |
|---|---|---|---|---|---|---|---|---|---|
| | ΔG (kJ/mol) | ΔH (kJ/mol) | ΔS J/(K*mol) | $K_d$ (µM) | n | % | p value* | % | p value* |
| FTH1-WT | −32.2 ± 1.1 | −23.0 ± 1.8 | 30.9 ± 8.2 | 2.4 ± 1.1 | 2.5 ± 0.6 | 100 | n.a. | 100 | n.a. |
| FTH1-A19Y | – | – | – | | – | 8.86 | <0.0001 | 0 | <0.0001 |
| FTH1-N22A | No Binding | | | | | 3.72 | <0.0001 | 0.89 | <0.0001 |
| FTH1-R23A | No Binding | | | | | 5.35 | <0.0001 | 7.04 | <0.0001 |
| FTH1-R23I | – | – | – | | – | 13.0 | <0.0001 | 23.7 | <0.0001 |
| FTH1-R23L | No Binding | | | | | 86.0 | 0.4614 | 33.7 | 0.0005 |
| FTH1-S114D | - 31.2 ± 0.3 | - 19.5 ± 2.1 | 39.1 ± 5.9 | 3.5 ± 0.5 | 3.0 ± 0.1 | 111 | 0.6897 | 23.9 | <0.0001 |
| FTH1-S114E | No Binding | | | | | 113 | 0.5198 | 24.9 | <0.0001 |
| FTL-WT | – | – | – | | – | 14.0 | <0.0001 | 0 | <0.0001 |
| L2H | −30.1 ± 0.3 | −22.0 ± 0.4 | 26.9 ± 2.4 | 5.6 ± 0.7 | 1.9 ± 0.1 | 147 | <0.0001 | 15.5 | <0.0001 |

*Dunnett's multiple comparison.

Source data are provided as a Source Data file.

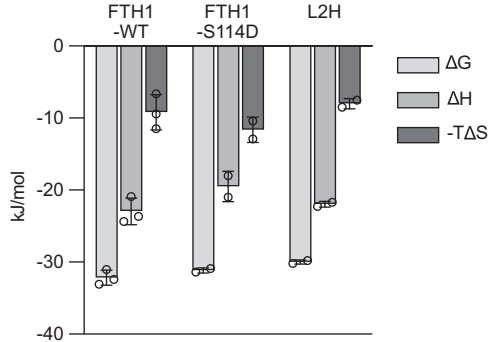

**Fig. 4 | Thermodynamic binding data from ITC measurements of NCOA4$^{FB}$-FTH1 and its mutants.** FTH1-*WT*, FTH1-S114D and L2H mutants show similar binding behaviors as apparent from their comparable binding parameters. The heat released in the ITC experiments of the other mutants was below the detection limit of the instrumentation (Supplementary Fig. 5). ΔG: Gibbs free energy changes, ΔH: binding enthalpy changes, and -TΔS: entropy changes of the reaction. Data are presented as mean values ± s.d. of technical duplicate or triplicate measurements (n = 2–3) from a representative experiment (of two to three biologically independent experiments with similar results). Source data are provided as a Source Data file.

S114D and S114E (Supplementary Fig. 3). Finally, comparing the amino acids of FTH1 and FTL at the NCOA4-FTH1 interaction interface, we recognized four amino acids in FTH1 that, if introduced into the same cleft in FTL, could result in establishing interaction between this FTL mutant and NCOA4. To test this hypothesis, we generated mutant FTL S19R, Q26Y, A86D, K106N (hereafter named L2H mutant), (Fig. 3d, and Supplementary Fig. 4).

The thermodynamics of NCOA4$^{FB}$ binding to FTH1 and its mutants were determined by isothermal titration calorimetry (ITC), which directly measures the heat of the binding reaction and the partitioning of the binding free energy into enthalpic and entropic components (Fig. 4, Supplementary Fig. 5, and Table 1). It also allows the identification of the nature and magnitude of the forces involved in the interaction. Overall, the ITC results agree with the structural data and emphasize the important role of hydrogen bonds in stabilizing the NCOA4-FTH1 interface. For instance, the mutant N22A which abolishes the only solvent-protected H-bonds had a dramatic effect on the heat of the binding reaction between NCOA4$^{FB}$ and FTH1, thus eliminating

the binding isotherm (Supplementary Fig. 5b). Similarly, while preserving the hydrophobic environment of the NCOA4$^{FB}$ binding pocket, the mutant R23L, like R23A, abolishes an H-bond with S492, leading to a dramatic decrease in the reaction binding enthalpy and a total loss of the binding isotherm (Supplementary Fig. 5c, d). On the other hand, the phosphomimetic mutant S114D showed a similar binding pattern to wild-type (*WT*)-FTH1 consistent with our cellular-based assays (see below, Fig. 5), but the homologous mutant S114E had undetectable binding heat (Fig. 4 and Supplementary Fig. 5e, f, and Table 1). This difference can be explained by the steric hindrance introduced by the larger glutamate residue that prevents the binding of NCOA4. Remarkably, the FTL mutant L2H demonstrated binding comparable to *WT*-FTH1 (Fig. 4, Supplementary Fig. 5g), thus revealing the structural basis for the specificity of NCOA4 towards FTH1.

## FTH1 mutants modulate intracellular NCOA4 binding and localization

We next examined the ability of FTH1 mutants to bind NCOA4 in cells. Because FTH1 forms 24-mer complexes in cells, we utilized CRISPR/Cas9-mediated genome editing to engineer *FTH1*-deficient (knockout: KO) clonal HEK293T and HeLa cell lines. *FTH1* KO clones were obtained and analyzed for FTH1 protein expression (Supplementary Fig. 6a). To determine whether mutant FTH1 binds NCOA4 in cells, we transiently expressed C-terminally-HA-FLAG-tagged NCOA4 and N-terminally MYC-tagged *WT* or mutant FTH1 in *FTH1*-KO HEK293T cells followed by MYC affinity purification to purify tagged FTH1. Consistent with previously published results, *WT*-FTH1 bound to NCOA4 but FTH1-R23A mutant abrogated binding to NCOA4 (Fig. 5a). FTH1-A19Y and FTH1-N22A abrogated binding to NCOA4 demonstrating the relevance of the NCOA4-FTH1 structural analysis in cells. As predicted, FTH1-R23I and FTH1-R23L mutants retained partial binding to NCOA4, highlighting the importance of hydrophobic side chain interactions in the FTH1 binding pocket. FTH1-S114D and FTH1-S114E phosphomimetic mutants demonstrated binding to NCOA4. We next examined the ability of the FTL mutant L2H to bind NCOA4 in cells. *WT*-FTL demonstrated no binding to NCOA4, as expected (Fig. 5b); however, L2H demonstrated binding to NCOA4.

To determine the colocalization of FTH1 and FTL mutants with NCOA4 in cells, we generated HeLa *FTH1*-KO cell lines stably expressing mRuby3-FTH1/FTL mutants with mGFP-NCOA4[9] (Supplementary Fig. 6b, c). With stable expression of mRuby3-FTH1-*WT* there was an increase in exogenous mGFP-NCOA4 levels suggesting that expression

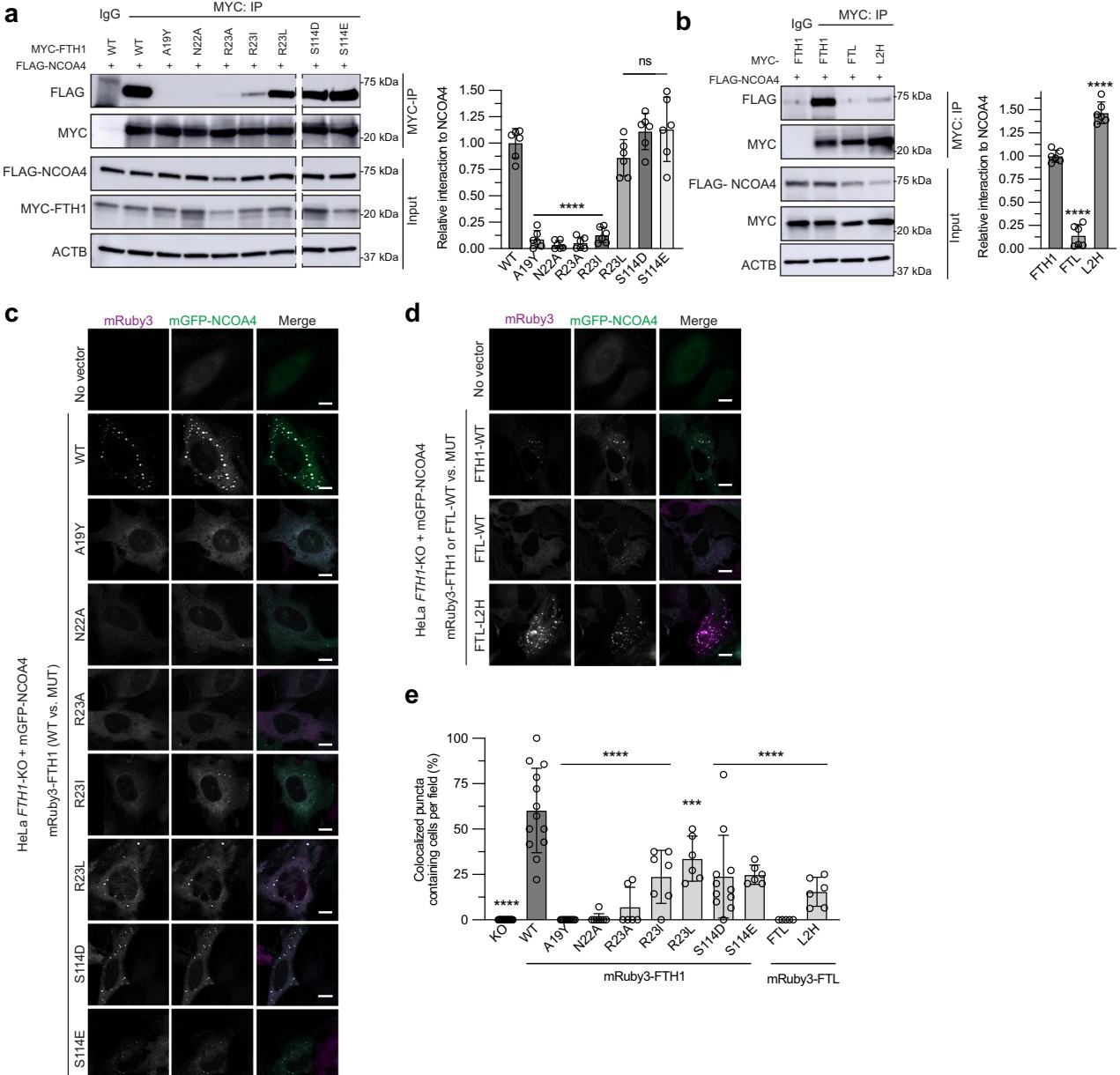

**Fig. 5 | FTH1 and FTL mutants modulate affinity and colocalization with NCOA4 in cells. a** Extracts from HEK293T-*FTH1* KO cells expressing the indicated proteins were immunoprecipitated with anti-MYC or IgG as negative control and immunoblotted with the indicated antibodies. Anti-ACTB, loading control. Relative quantification of FTH1:NCOA4 interaction (normalized to immunoprecipitated FTH1 levels). Data are presented as mean relative interaction ± s.d. of two biologically independent experiments each with technical triplicate measurements, *n* = 6. Differences between the *WT* and Mutant interactions were analyzed statistically using one-way ANOVA with Dunnett's multiple comparisons test. ****$p < 0.0001$ (A19Y, N22A, R23A, R23I), ns non-significant (R23L: $p = 0.4614$, S114D: $p = 0.6897$, S114E: $p = 0.5198$). **b** Extracts from HeLa-*FTH1* KO cells expressing the indicated proteins were immunoprecipitated with anti-MYC and immunoblotted with the indicated antibodies. Anti-ACTB, loading control. Relative quantification of FTL:NCOA4 interaction (normalized to immunoprecipitated FTH1/FTL levels and NCOA4 input levels). Data are presented as mean relative interaction ± s.d. of two biologically independent experiments each with technical triplicate measurements, *n* = 6. Differences between the FTH1-*WT* and FTL or L2H were analyzed statistically using one-way ANOVA with Dunnett's multiple comparisons test. ****$p < 0.0001$ (FTL, L2H). **c** *WT* or mutants of mRuby3-FTH1 were stably expressed in HeLa-*FTH1* KO cells harboring mGFP-NCOA4, cells were fixed and visualized for mRuby3-FTH1

puncta and mGFP-NCOA4 co-localization by fluorescence microscopy. Given the wide degree of variability in mGFP-NCOA4 expression level (Supplementary Fig. 6b, c, discussed in text), the maximum threshold of the mGFP-NCOA4 channel in images presented in Fig. 5c, d is adjusted according to the mGFP-NCOA4 expression level to allow for visualization of any colocalization of lower expressed mGFP-NCOA4 with FTH1 mutants. Thresholding for the mRuby-FTH1/FTL channel is consistent across all experiments. **d** mRuby3-FTH1-*WT* or FTL-*WT* and FTL-L2H mutant were stably expressed in HeLa-*FTH1* KO cells harboring mGFP-NCOA4, cells were fixed and visualized for mRuby3 puncta and mGFP-NCOA4 co-localization by fluorescence microscopy. **e** Percent of cells with colocalized mRuby3-FTH1/FTL:mGFP-NCOA4 puncta. At least 5 fields and *n* > 50 cells total were quantified per condition (KO: *n* = 98, WT: *n* = 147, A19Y: *n* = 151, N22A: *n* = 111, R23A: 74, R23I: *n* = 71, R23L: *n* = 76, S114D: *n* = 118, S114E, *n* = 57, FTL: *n* = 118, L2H: *n* = 98). Data are presented as mean percentage ± s.d. (each data point represents the percent colocalized puncta per an individual field). Differences among the cells expressing the *WT* versus mutants were analyzed statistically using one-way ANOVA with Dunnett's multiple comparisons test. ***$p = 0.0005$ (R23L), ****$p < 0.0001$ (KO, A19Y, N22A, R23A, R23I, S114D, S114E, FTL, L2H). The scalebar is 10 µm in all micrographs. Source data are provided as a Source Data file.

of a version of FTH1 capable of binding to NCOA4 stabilized NCOA4 (potentially by sequestering NCOA4 from HERC2-mediated degradation[10]) (Supplementary Fig. 6b). Conversely, the expression of mRuby3-FTH1-R23A that is incapable of binding NCOA4 did not demonstrate stabilization of NCOA4. Indeed, the expression of FTH1 mutants incapable of binding NCOA4 in cellular immunoprecipitation experiments (FTH1-A19Y and -N22A) demonstrated no increase in NCOA4 levels above the parental FTH1 knockout cell line. In contrast, the expression of FTH1-R23I, R23L, S114D, and S114E mutants capable of binding NCOA4 in cells, demonstrated variable increases in cellular NCOA4 levels. Finally, the expression of the L2H mutant demonstrated an increase in NCOA4 levels (Supplementary Fig. 6b, c). Furthermore, anti-GFP immunoprecipitation of mGFP-NCOA4 demonstrated concordant interactions with mRuby-FTH1/FTL constructs (Supplementary Fig. 6d, e) in comparison to results with anti-MYC immunoprecipitation of myc-FTH1/FTL (Fig. 5a, b).

We next analyzed the colocalization of mRuby3-FTH1 with mGFP-NCOA4. Recent studies have shown that NCOA4 and FTH1 form cytoplasmic condensates that are autophagocytosed for delivery to lysosomes[9,12]. In FTH1-KO cells expressing mRuby3-FTH1-*WT* and mGFP-NCOA4, mRuby3-FTH1 formed puncta with extensive colocalization with mGFP-NCOA4 (Fig. 5c, e). Co-expression of mRuby3-FTH1-R23A and mGFP-NCOA4 demonstrated diffuse mRuby3-FTH1-R23A signal, as previously demonstrated[10,12]. Similarly, mRuby3-FTH1-A19Y and -N22A demonstrated diffuse mRuby3 signal with no colocalization with mGFP-NCOA4. R23I, R23L, S114D, and S114E FTH1 mutants demonstrated an intermediate colocalization of mRuby-FTH1 punctate structures consistent with intermediate binding to NCOA4 (Fig. 5c, e).

As expected, mRuby3-FTL-*WT* demonstrated a diffuse signal for mRuby3 (Fig. 5d). Remarkably, mRuby3-FTL-L2H mutant formed punctate structures colocalized with mGFP-NCOA4, indicating a new ability for the mutant FTL to interact with and colocalize with NCOA4 (Fig. 5d, e).

## Discussion

In this research, we have determined the structural basis for the interaction of NCOA4 and FTH1. Our results show that the interaction between NCOA4 and FTH1 is localized to a single subunit of FTH1. Thus, the previously observed 1:3 stoichiometric ratio of the NCOA4$^{FB}$·FTH1 complex[11] cannot be explained by the direct binding of one NCOA4 chain to several FTH1 subunits, suggesting that this ratio results from steric hindrance between the intrinsically disordered regions of the NCOA4$^{FB}$ fragment (Fig. 1a). The stoichiometric ratio of full-length NCOA4 with FTH1 is unknown and similarly whether ferritin complexes interact with multiple NCOA4 molecules in cells is unknown. However, it has been shown that dimerization/oligomerization of NCOA4 is required for condensate formation and ferritin degradation[9]. Determining the arrangement and ratio of full-length NCOA4 with FTH1 homopolymers and heteropolymers with different ratios of FTH1 and FTL under various iron-loading conditions will be critical for understanding the regulation of NCOA4·FTH1 complexes in vivo.

Our analysis revealed the role of several structural features on the FTH1 surface that are crucial for the formation of the NCOA4·FTH1 complex. The binding interface between NCOA4 and FTH1 is predominantly stabilized by hydrophobic interactions, where surface-exposed hydrophobic residues in FTH1 form a series of three back-to-back binding pockets that accept three highly hydrophobic residues from NCOA4. The thermodynamic signatures of the interaction between NCOA4$^{FB}$·FTH1 indicate both enthalpically and entropically driven binding reactions. The large favorable enthalpy changes reflect the relatively strong contribution of H-bonds to the formation of the NCOA4$^{FB}$·FTH1 complex, while the positive entropic contribution (i.e., positive ΔS) illustrate the hydrophobicity of the binding pocket and changes in the hydration of the two proteins and the release of water

molecules upon binding (Fig. 4 and Supplementary Fig. 5). This favorable entropy of solvation is the predominant force associated with the binding energy of hydrophobic groups and is reflected in both the ITC results and the NCOA4 immunoprecipitation and colocalization experiments (Fig. 5a, e), highlighting the important contributions of hydrophobic interactions to the formation of the NCOA4 and FTH1 complex. We evaluated the effect of the phosphomimetic mutations (S114D and S114E) on the NCOA4-FTH1 interaction, following the indication that this site is phosphorylated. The S114D mutation had no effect, whereas S114E disrupted the interaction in ITC but not in the cellular context. This discrepancy may be explained by the differences between the truncated NCOA4$^{FB}$ used in the ITC measurements versus the full-length NCOA4 used in cellular studies. Full-length NCOA4 is known to oligomerize via its N-terminal coiled-coil domain that is absent in the NCOA4$^{FB}$; this may produce avidity effects that facilitate binding of the S114E mutant that is not measured by the ITC measurements. Regardless, in our cellular experimental set-ups, both phosphomimetic mutations of S114 do not appear to affect the NCOA4-FTH1 interaction, suggesting that S114 phosphorylation is not a major modulator of the NCOA4-FTH1 interaction.

NCOA4-mediated trafficking of ferritin is a major mechanism controlling the level of free iron in the cell[3]. Physiologically, the pathway is present at basal levels and is further activated under conditions of iron starvation to mobilize iron from ferritin to maintain iron levels[15,16]. Conversely, in acute iron loading, this pathway is blocked to prioritize the recruitment of iron from the cytosol into ferritin and avoid inappropriate ferritin degradation[10,12]. Consequently, it is reasonable to presume the existence of an iron- or redox-dependent control layer that directly regulates the NCOA4-FTH1 interaction, in addition to already known indirect regulation mechanisms, such as HERC2-mediated degradation of NCOA4[10] and NCOA4 condensation[9,12].

Based on our structural analysis, we designed mutations of FTH1 that modulate its interaction with NCOA4 and an FTL mutant capable of binding NCOA4, as demonstrated by ITC measurements (Fig. 4 and Supplementary Fig. 5), and in cellular immunoprecipitation assays and colocalization experiments (Fig. 5 and Supplementary Fig. 6). Thus, the binding site described here can entirely explain the NCOA4-FTH1 interactions. This was demonstrated by the loss of affinity and function induced by our structure-predicted point mutations of amino acids of the binding site (A19Y, N22A, and R23A), and by the gain of function L2H mutant of FTL. None of the interactions stabilizing the NCOA4-FTH1 interface are expected to be affected by iron availability or the redox potential of the cell. Thus, our finding explains the basal level of ferritinophagy present in all cells.

Numerous pathologies are associated with the dysregulation of cellular iron levels. The shift out of balance of the cytosolic iron regulation system can increase sensitivity to multiple forms of cell death, including ferroptosis, an iron-dependent lipid peroxidation-induced regulated cell death mechanism. For instance, premature neuronal death[17], heart failure[18], and ischemia/reperfusion-induced tissue damage[19] can all result from ferroptosis. As NCOA4-mediated ferritinophagy controls cellular free iron levels, modulation of the pathway influences sensitivity to induction of ferroptosis. Indeed, down-regulation of NCOA4 decreased sensitivity to ferroptosis induction and NCOA4 upregulation was associated with increased sensitivity to ferroptosis induction[18,20,21]. NCOA4-mediated ferritinophagy has also been hijacked in pathophysiological conditions to provide iron to proliferating cells, e.g., in pancreatic cancer[22,23], and infections[24]. For these reasons, blocking the interaction between NCOA4 and FTH1 using small molecules is considered a promising therapeutic strategy for numerous pathologies. With no structures, up until now, the development of such compounds was led by observational rather than rational search, and the resulting leads seem to interact with NCOA4[25]. We expect that the intricate binding site of NCOA4 on the surface of

FTH1 described in this research could serve as a target for the rational design of molecules blocking NCOA4·FTH1 complex formation.

## Methods

### Expression and purification of human FTH1, and its mutant

The plasmid bearing *WT*-FTH1 was obtained from Addgene (#122652), and all mutants were ordered from Twist Bioscience using the pET-29b(+) plasmid as a backbone. *E. coli BL21* (DE3) harboring the plasmid for *WT*-FTH1 or its mutants (as indicated in the text) were grown in LB to an OD 600 of 0.8 at 37 °C. Induction was initiated by the addition of 1 mM isopropyl-β-d-1-thiogalactopyranoside (IPTG), and the culture was grown at 20 °C overnight. Harvested cells were stored at -80 °C until used.

For purification, cells were resuspended in lysis buffer (25 mM Tris-Cl pH 7.5, 150 mM NaCl, 3 mM PMSF) and lysed by passing the suspension three times in an Avestin Emulsiflex C5 homogenizer. After removing cell debris by centrifugation at 4700 × *g* over 15 min, the supernatant was aliquoted (1 ml) and incubated at 55 °C for 10 min. Aliquots were cooled down in ice water for 5 min and centrifuged at 20,000×g for 10 min. The supernatants were combined, the salt content was supplemented to final concentrations of 20 mM $MgCl_2$ and 10 mM $CaCl_2$, and the mixture was treated with DNase I (Merck 10104159001) for 1.5 h at 37 °C. After DNase treatment, the mixture was heat treated once more, by heating to 75 °C for 10 minutes and a cooldown in ice water for 5 minutes, followed by precipitation of denatured proteins by centrifugation at 20,000 × *g* for 10 min. The supernatant was diluted six times by the addition of a low salt dilution buffer (10 mM Tris-HCl pH 7.5, 10 mM NaCl) and loaded onto HiTrap Q HP, and the column was washed with the low salt buffer until the OD 280 fell back to the background level. The protein was eluted by a gradient running up to 300 mM NaCl, the main elution peak is readily apparent in the OD 280 signal, and its location depends on the ferritin mutant being purified. Fractions were analyzed by SDS-PAGE and were selected according to their purity. To concentrate the selected fractions, they were supplemented with a fine powder of ammonium sulfate at a ratio of 0.52 gm powder for every ml of eluted protein, followed by a precipitation step of the target protein by 50,000×g centrifugation over 20 min. The pellet was resuspended in 25 mM Tris–HCl pH 7.5 150 mM NaCl and the suspension was desalted by PD10 equilibrated by the same buffer to remove the remaining ammonium sulfate. The purity and assembly of the resulting protein were confirmed by SDS-PAGE and native PAGE. Aliquots were flash-frozen into liquid nitrogen and kept at -80 °C until use.

### Expression and purification of NCOA4^FB

*E. coli T7 Lys* cells, harboring the plasmid for the His-tagged NCOA4^FB (aa383-522), were cultured in LB medium at 37 °C until reaching an OD600 of 0.8. Induction was initiated by the addition of 1 mM isopropyl-β-d-1-thiogalactopyranoside (IPTG), and the culture was further incubated at 30 °C for 3 hours. Once harvested, cells were stored at −80 °C until needed.

Cells were resuspended in lysis buffer (50 mM Tris pH 7.5, 150 mM NaCl, 3 mM TCEP) and lysed by passing the suspension three times in an Avestin Emulsiflex C5 homogenizer. Post lysis, the mixture was centrifuged at 50,000 × *g*. Subsequently, the clarified supernatant was loaded onto a HisTrap column. The column underwent several washes with increasing concentrations of imidazole: 10 mM, 20 mM, and 40 mM. The protein was then eluted in one step with a buffer containing 250 mM imidazole. NOCA4^FB-containing fractions had a dark reddish-brown color. The fractions were pooled and subjected to a buffer exchange using a PD10 desalting column equilibrated with thoroughly degassed buffer 'standard' buffer (25 mM Tris pH 7.5, 150 mM NaCl). Protein destined for SEC analysis and for Cryo-EM sample preparation was supplemented by 25 mM of TCEP and was used fresh.

### Cryo-EM gird preparation, data collection, and image processing

Cryo-EM grids were plunged frozen into liquid ethane, using a home-built device. -1 µl of FTH1 were placed on the grid, these were supplemented with ~2 µl of NCOA4^FB to a final ratio of ~24:24 and a concentration of ~0.5 mg/ml. The grids were manually blotted and immediately plunged frozen and kept under liquid nitrogen until use.

Single particle Cryo-EM datasets were collected under cryogenic conditions at the BGU's Cryo-EM core facility using a Glacios Cryo-TEM equipped with a Falcon 4E direct electron detector placed behind a Selectris X imaging energy filter. Micrographs were collected at a calibrated pixel size of 0.89 Å at a total dose of 30 $e^-/Å^2$ and a defocus range of −0.5 to −2.0 µm. Data collection and microscope operation were managed by EPU (Version 3.4.0). Altogether 905 micrographs deemed with acceptable quality were collected, as was assessed by the extent of their CTF ring and manual observation, ruling out micrographs with excess carbon area or contaminations.

Data was processed by RELION3 (version: 3.1.1-commit-879ea4)[26], assuming both C1 and O symmetries (Supplementary Fig. 1 and Table 1 summarize the image processing parameters and validation). The contrast transfer function of each micrograph was estimated by CTFFind4 (Version 4.1.14). Symmetry relaxation, and symmetry expansion followed by focused refinement did not provide regions with clearer NCOA4 chains. For that reason, the 3D-EM map used for model building was the map generated by assuming 'O' symmetry reaching a resolution of ~2.9 Å, as assessed by the 'gold-standard' FSC plot (Supplementary Fig. 1). The molecular model of NCAOA4^FB·FTH1 was built into the map using coot[27,28], and was subjected to real space refinement by PHENIX[29]. Model building started from the already published molecular model of human FTH1[30] (PDB 7RRP).

### Isothermal Titration Calorimetry (ITC)

ITC experiments were carried out on a TA Instruments small-volume Nano ITC instrument with ITCRun (v3.8.0.18730) data collection software with an active cell volume of 185 µL. While the association constant (Ka), stoichiometry (n), and enthalpy change (ΔH°) of binding are directly determined from the ITC binding curve, the Gibbs free energy of binding (ΔG°) and entropy change (ΔS°) of the reaction were calculated using the relationships ΔG° = −RT ln Ka and ΔG° = ΔH° − TΔS°. All ITC titrations were performed at 25.00 °C in 100 mM MOPS and 100 mM NaCl, pH 7.4 using an automated sequence of 16 injections, 3 µL each of NCOA4 into the ITC sample cell containing ferritin. ITC data were analyzed with the TA Instruments' NanoAnalyze software (v3.12.5) using one class of independent binding sites after a background correction in the absence of ferritin to account for the heat of mixing and heat of dilution. All experiments were repeated two or three times to ensure reproducibility with standard errors from replicate determinations reported for the experimental thermodynamic values. Raw data and fits are available in Supplementary Fig. 5 and the Source Data.

### Mammalian cell culture

HEK293T and HeLa cells were obtained from ATCC. Cell lines were maintained in a centralized cell bank, authenticated by assessment of cell morphology as well as short tandem repeat fingerprinting, and routinely inspected for *Mycoplasma* contamination using PCR (most recently in June 2023 with all cells testing negative). After thawing, cell lines were cultured for no longer than 45 days. Cell lines were maintained at 37 °C with 5% $CO_2$ and grown in DMEM supplemented with 10% FBS and 1% penicillin/streptomycin.

### CRISPR–SpCas9 genome editing in cell lines

Clustered Regularly Interspaced Short Palindromic Repeats (CRISPR)/Cas9-mediated genome editing was used to generate clonal *FTH1* knockout HEK293T and HeLa cell lines. A crRNA against *FTH1* (5'ACCATGGACAGGTAAACGT 3') was designed and obtained using the

IDT crRNA design tool. Equimolar amounts (240 pmoles) of the crRNA and tracrRNA (IDT #1072532) along with Cas9 (IDT #1081058) were incubated to form Cas9 RNP complex. Cas9 RNP complexes with electroporation enhancer (IDT #1075915) were electroporated using a Lonza 4D-Nucleofector as per the manufacturer's guidelines. Single-cell cloning was carried out using the limited dilution method and *FTH1* knockout clones were identified by western blot using antibodies whose epitopes were distinct from Cas9 RNP target regions.

### cDNA expression constructs

*FTH1* (NM_002032.3) and *FTL* (NM_000146.4) wild type and mutants with a 5′, in frame, MYC tag were synthesized by Twist Bioscience and inserted in the pTwist EF1 Alpha vector. Constructs were expressed in the indicated cell lines by transient transfection using PolyJet™ (SignaGen Laboratories, SL100688). pMRX-IP-mGFP-NCOA4 (Addgene plasmid #192802), pMRX-IB-mRuby3-FTH1 (Addgene plasmid #192804), and pMRX-IB-mRuby3-FTL (Addgene plasmid # 192805) plasmids were gifts from Noboru Mizushima. FTH1 mutants, and L2H mutant were subcloned into pMRX-IB-mRuby3-FTH1 or pMRX-IB-mRuby3-FTL vector using Gibson Assembly Cloning Kit (NEB, E5510S).

### Stable expression in HeLa cells by retrovirus transduction

For the preparation of the retrovirus solution, Plat-E cells (Cell BioLabs, RV-101) were transfected with the pMRX-IP-mGFP-NCOA4 or pMRX-IB-based retroviral plasmids together with gag/pol and VSV-G using PolyJet™ (SignaGen Laboratories, SL100688). The retrovirus-containing medium was collected and filtered through a 0.45 μm filter unit (Corning, 431220) and added to HeLa-*FTH1* KO cells with 8 μg/ml polybrene (H9268; Sigma-Aldrich). The transduced HeLa cells were then selected with 2 μg/mL puromycin (Santa Cruz, sc-108071B) or 5 ug/mL blasticidin (GoldBio, B-800-100) for 2-7 days. mGFP- and mRuby3-tagged exogenous proteins were confirmed by western blot and fluorescence microscopy.

### Western blotting

Cells were lysed in RIPA buffer with protease inhibitors, centrifuged at 21,000 x g, and supernatants were collected. Protein (60 μg) was resolved on 4% to 20% SDS-PAGE gels and transferred to nitrocellulose or PVDF membranes. Membranes were blocked in 5% milk and incubated with primary antibodies and then with peroxidase-conjugated secondary antibody. Membranes were developed using the ECL Detection System (Thermo, 32209). The following antibodies were used: NCOA4 (Santa Cruz Biotechnology, sc-373739, 1:100), FTH1 (Cell Signaling Technology (CST), #4393, 1:2,000), ACTB (Sigma, A5441, 1:5,000), IREBP2/IRP2 (CST, #37135, 1:100), anti-DYKDDDDK (CST, #14793, 1:1000); HA-Tag (CST, #3724, 1:1000), c-Myc (Invitrogen, MA1-980, 1:1000), GFP (CST, #2956, 1:1000) Secondary antibodies were as follows: anti-rabbit IgG (H + L) HRP conjugate (Thermo, 31460, 1:5000), anti-mouse IgG (H1L) HRP conjugate (Promega, W4021, 1:5000), and anti-mouse IgG Fc cross-absorbed HRP conjugate (Invitrogen, 31439, 1:5000). All uncropped western blot membranes are depicted in Source Data (for Fig. 5 blots) and Supplementary Fig. 7 (for Supplementary Fig. 6 blots).

### Immunoprecipitation

For MYC affinity purification of MYC-FTH1/FTL wild type or mutant variants, HEK293T or HeLa cells were harvested at ~80% confluency and lysed in 50 mM Tris−HCl (pH 7.5), 150 mM NaCl, 0.5% Nonidet P40, 1 mM DTT and protease inhibitors (Roche). Cleared extracts were subjected to IP with anti-MYC agarose (Thermo Scientific, 20169). Complexes were washed with lysis buffer and subjected to SDS-PAGE and immunoblot with the indicated antibodies. For GFP affinity purification of mGFP-NCOA4, HeLa cells expressing the indicated constructs were harvested, lysed, and subjected to anti-GFP IP as described above. Immunoblots were performed as above.

### Immunofluorescence

To examine mGFP-NCOA4 and mRuby3-FTH1 / FTL localization, cells were plated on glass coverslips (Fisher, 1254583) and fixed with 4% paraformaldehyde for 15 minutes at room temperature. Coverslips were mounted using ProLong™ Glass Antifade mountant (Invitrogen, P36980) and allowed to dry overnight. Cells were imaged using a Zeiss AxioObserver using a 63x objective using the same exposure times across samples. Image processing and quantification were performed using ImageJ software v1.54 h. Given the wide degree of variability in mGFP-NCOA4 expression level (Supplementary Fig. 6b, c, discussed in results), the maximum threshold of the mGFP-NCOA4 channel in images presented in Fig. 5c, d is customized for each mutant according to expression level to allow for detection of any colocalization with mRuby-FTH1, even at a lower intensity due to lower expression levels of mGFP-NCOA4. Thresholding for the mRuby-FTH1/FTL channel is consistent across all experiments.

### Statistical analysis

For all data, statistical analysis was done using GraphPad PRISM v10.2.0. No statistical methods were used to predetermine sample size. For multiple comparisons, one-way ANOVA with Dunnett's multiple comparisons tests were performed.

### Reporting summary

Further information on research design is available in the Nature Portfolio Reporting Summary linked to this article.

## Data availability

The Cryo-EM map and resulting molecular model are available in the Electron Microscopy Data Bank (EMDB) accession number EMD-18658 and the Protein Data Bank (PDB) accession code 8QU9, respectively. The raw ITC data and western blots data are provided in the supplementary material and source data. The optical microscopy data is available on figshare: https://doi.org/10.6084/m9.figshare.25526188. Source data are provided with this paper.

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

## Acknowledgements

This research was supported by NSF-BSF collaborative grant 2231900 2022614 to GAF, EGMH, JDM, and FBA, the ISF personal grant 364/20 to GAF, and NIH grant R01 DK124384 and the Morris Family Pancreatic Cancer Research Fund to JDM. pMRX-IP-mGFP-NCOA4 (Addgene plasmid #192802), pMRX-IB-mRuby3-FTH1 (Addgene plasmid #192804), and pMRX-IB-mRuby3-FTL (Addgene plasmid # 192805) plasmids were gifts from Noboru Mizushima. We thank Prof. Raz Zarivach for his assistance in validating the molecular model, Mr. Dimitry Mayzus for his superb facility maintenance support, and Mr. Ajami Gikandi for assistance in generating the HeLa and HEK293T knockout cells. The authors are grateful for the generous support from the Guzik Foundation to BGU's Cryo-electron microscopy unit.

## Author contributions

G.A.F., J.D.M., F.B.A., and E.G.M.H. conceived the project, planned the experiments, and wrote the paper. F.H., G.A.F., R.Z., and A.S., collected, analyzed, and validated the cryo-EM data and the resulting structure. F.H., E.K., S.C.S., M.B., and AKS, purified the proteins. F.B.A., M.B., and A.K.S. performed and analyzed the ITC experiments. T.T.P.N., E.Y.K., and J.D.M. performed and analyzed the cellular assays.

## Competing interests

GAF and JDM hold the pending provisional patent application 63/534,363 for modulating ferritinophagy based on the structure and mutants described in this paper. JDM reports research support from Novartis and Casma Therapeutics and has consulted for Third Rock Ventures and Skyhawk Therapeutics, all unrelated to the submitted work. All other authors declare no conflict of interest.
