## [Peer Review File · Nature Communications]

Structural basis for the intracellular regulation of ferritin degradationREVIEWER COMMENTS

Reviewer #1 (Remarks to the Author):

This manuscript reports a comprehensive, structure-guided identification and validation study of NCOA4 and ferritin interactions. Key residues on both NCOA4 and FTH1 proteins are identified/validated from cryo-EM near-atomic resolution structure/model, mutagenesis, in vitro binding assays, pull-down assays, and cellular colocalization assays. These results will enhance the understanding of ferritinophagy and iron homeostasis. I support the publication of this manuscript after addressing the comments below in the revision.

Are these two results for S114D and S114E inconsistent? “the phosphomimetic mutant S114D showed a similar binding pattern to wild-type (WT)-FTH1 but the homologous mutant S114E introduced a steric hindrance that prevented the binding of NCOA4” (based on Fig. S4), and then in Fig. 4, the S114D/S114E results suggest both bind equally well.

According to the method section, it appears that 1:1 molar ratio was used for cryo-EM of the NCOA4/FTH complex. Is this 1:1 molar ratio (instead of more NCOA4) the major reason for “The overall density of the NCOA4FB segments in the 3D-EM map was lower than the density of the FTH1 region”? The authors should already have the data (ITC results) to estimate the binding affinity from which the binding occupancy for this 1:1 ratio condition could be estimated.

The FSC and the resolutions are dominated by the FTH cage (irrelevant to this study) instead of NCOA4 and the interface region (critical for this study). It will be useful to show the local resolution map highlighting the NCOA4 and the interface region and properly cite the resolutions for this region.

It would be useful to include a zoom-in view of a region of the density map in Fig 1 where the high-resolution densities (instead of the low-res cartoonish display in current Fig. 1) of both NCOA4 and FTH densities are clearly visible.

Fig. 1 needs to be improved to better illustrate the interactions between FTH and NCOA4 to better support the stated interactions in the last paragraph of the “The structure of the NCOA4-FTH1 complex” section.

Are FTH R23 and NCOA4 W497 engaged in a cation- π interaction?

Current writing might mislead the readers to believe the entire NCOA4 protein was solved in the cryoEM structure. The authors should make it clear in the revision where appropriate (especially in the abstract) that a NCOA4 small fragment was used and only a short 16 aa peptide was resolved. It might be useful to include a supplementary figure to depict the major domain/regions of the NCOA4 protein, fragment used in this study, and the resolved 16aa region. In this figure, the author could also consider including an AlphaFold model to show the intrinsically disordered nature.

Typo "dimmed". Should be "deemed"?

"Relion" should be changed to RELION

Reviewer #2 (Remarks to the Author):

In this manuscript the authors describe the structural basis for interactions between ferritin H and NCOA4 using cryo-EM, site-directed mutagenesis, ITC, and cell-based pull-down and colocalization studies. The authors report that a fragment of NCOA4 binds to FtnH along a groove between the first alpha helix (A) and the B-C loop on the surface of a single protomer. Bind is stabilized by a group of hydrophobic interactions between FtnH and the NCOA4 fragment. Site-directed mutations in Ftn H predicted to disrupt these interactions resulted in loss of binding activity as measured in vitro by ITC and in cells by co-localization and pull-down. Mutation of corresponding residues in FtnL, which does not bind NCOA4, to mimic Ftn H resulted in partial NCOA4 binding activity as measured by ITC and cell-based studies. The data are clear, the quality of the studies appears to be high, and the results fully support the authors' conclusions. This reviewer has some minor concerns and suggestions for improvement.

1. The authors use the term "gorge" to describe the region of ferritin that interacts with NCOA4. This reviewer was confused by this term as it has not been used to describe ferritin or other protein structures. Other terms that might be clearer could be "cleft," "furrow" or "groove".

2. Would an image of the FTN-NCOA4 structure rotated so as to position NCOA4 on top of the FtnH be useful? This could be added to Figure 1.

3. The authors should put results in the main text and not in the Figure legends. The legend for Fig. 2 reads like results. Limit the legend to simply describing the possible position and interactions of the side chains and make clear these are models only, not structures.

4. Negative data are very important for the interpretation of the Ftn mutants. Include a table that reports the ITC data for all of the mutants in the main text of the paper, not just in the supplement. Can you also report dissociation constants? Cell biologists are more familiar with this kinetic measurement.

5. The authors report that the phosphomimetic mutants of FtnH S114 are only partially active for binding. Yet the data in Fig. 4 indicate that these mutants are not different from wild type FtnH in their binding activity and therefore suggest phosphorylation does not modulate this interaction and would not warrant further study. Why are the authors suggesting otherwise?

6. Text in the ITC images in supplement S4 are too small to read. Annoying.

7. Co-localization experiments are very good.

8. Discussion makes clear that this structure does not explain the 1:3 stoichiometry of Ftn-NCOA4 binding. The authors suggest that this is likely due to steric hindrance from disordered regions of the NCOA4 protomer. Can you model this at all?

Reviewer 1

We thank the reviewer for the supportive and meticulous review of our manuscript. We agree with most of the comments raised by the reviewer. Detailed responses and a list of changes we introduced to the text, in accordance with the review, are listed below each of the reviewer's comments.

Point 1:

Are these two results for S114D and S114E inconsistent? “the phosphomimetic mutant S114D showed a similar binding pattern to wild-type (WT)-FTH1 but the homologous mutant S114E introduced a steric hindrance that prevented the binding of NCOA4” (based on Fig. S4), and then in Fig. 4, the S114D/S114E results suggest both bind equally well.

Answer 1: Thank you for pointing out this confusing description of our results. This clarity issue results from the convergence of several points of critique raised by both reviews. The S114D and S114E mutants are a case where there is an inconsistency between the cellular assays and the ITC measurements. We addressed this point by introducing several changes to the text:

a. The main text was rephrased in several places:

Original:

On the other hand, the phosphomimetic mutant S114D showed a similar binding pattern to wild-type (WT)-FTH1 but the homologous mutant S114E introduced a steric hindrance that prevented the binding of NCOA4.

Now reads:

On the other hand, the phosphomimetic mutant S114D showed a similar binding pattern to wild-type (WT)-FTH1 **consistent with our cellular-based assays (Fig. 5), but the homologous mutant S114E had undetectable binding heat (Fig. 4 and Supplementary Fig. 4d-e). This difference can be explained by the steric hindrance introduced by the larger glutamate residue** that prevents the binding of NCOA4^{FB}.

Original:

As predicted, FTH1-R23I and FTH1-R23L mutants retained partial binding to NCOA4 highlighting the importance of hydrophobic side chain interactions in the FTH1 binding pocket. FTH1-S114D and FTH1-S114E phosphomimetic mutants demonstrated partial binding to NCOA4 suggesting that S114 phosphorylation observed in high throughput phosphoproteomic experiments may modulate the NCOA4-FTH1 interaction and therefore warrants further study. We next examined the ability of the FTL mutant L2H to bind NCOA4 in cells.

Now reads:

As predicted, FTH1-R23I and FTH1-R23L mutants retained partial binding to NCOA4, highlighting the importance of hydrophobic side chain interactions in the FTH1 binding pocket. FTH1-S114D and FTH1-S114E phosphomimetic mutants demonstrated binding to NCOA4. We next examined the ability of the FTL mutant L2H to bind NCOA4 in cells.

We added the following paragraph to the discussion:

We evaluated the effect of the phosphomimetic mutations (S114D and S114E) on the NCOA4-FTH1 interaction, following the indication that this site is phosphorylated. The S114D mutation had no effect, whereas S114E disrupted the interaction in ITC but had no effect in the cellular context. This discrepancy may result from the differences between the truncated NCOA4^{FB} used in the ITC measurements versus the full-length NCOA4 used in cellular studies. Full-length NCOA4 oligomerizes via its N-terminal coiled-

coil domain that is absent in the NCOA4^{FB}; this may produce avidity effects that facilitate binding of the S114E mutant not detected by the ITC measurements. Regardless, in our cellular experimental set-ups, both phosphomimetic mutations of S114 do not affect the NCOA4-FTH1 interaction, suggesting that S114 phosphorylation is not a major modulator of the NCOA4-FTH1 interaction.

- b. A table allowing a clear comparison between the different binding assays was moved to the result section.

Table 1: Comparison of the affinity measurements between NCOA4 and FTH1-WT and its mutants obtained by ITC, immunoprecipitation and colocalization assays.

Point 2:

According to the method section, it appears that 1:1 molar ratio was used for cryo-EM of the NCOA4/FTH complex. Is this 1:1 molar ratio (instead of more NCOA4) the major reason for “The overall density of the NCOA4^{FB} segments in the 3D-EM map was lower than the density of the FTH1 region”? The authors should already have the data (ITC results) to estimate the binding affinity from which the binding occupancy for this 1:1 ratio condition could be estimated.

Answer 2: Based on our previous experience with the interaction of NCOA4^{FB} and FTH1¹ and in accordance with our current ITC measurements, at saturation, the stoichiometry of the NCOA4^{FB}-FTH1 complex is 8 NCOA4^{FB} chains per 24 FTH1 subunits (1:3 ratio). Thus, the 1:1 molar ratio we used in the cryo-EM sample preparation is a large excess of NCOA4^{FB}. These conditions were selected to ensure the highest occupancy possible of NCOA4^{FB}-FTH1 complexes. As the reviewer suggests, we explain the lower density of the NCOA4^{FB} by this lower occupancy that is inherent to the system. We describe this point at the beginning of the results section:

The overall density of the NCOA4^{FB} segments in the 3D-EM map was lower than the density of the FTH1 region. This can be explained by NCOA4^{FB}:FTH1-subunits occupancy ratio of ~1:3 found in previous studies.

Point 3:

The FSC and the resolutions are dominated by the FTH cage (irrelevant to this study) instead of NCOA4 and the interface region (critical for this study). It will be useful to show the local resolution map highlighting the NCOA4 and the interface region and properly cite the resolutions for this region.

Answer 3: We thank the reviewer for this point, a local resolution map focusing on the resolution of the NCOA4^{FB} chain was added to Supplementary Fig. 1.

Point 4:

It would be useful to include a zoom-in view of a region of the density map in Fig 1 where the high-resolution densities (instead of the low-res cartoonish display in current Fig. 1) of both NCOA4 and FTH densities are clearly visible.

Fig. 1 needs to be improved to better illustrate the interactions between FTH and NCOA4 to better support the stated interactions in the last paragraph of the “The structure of the NCOA4-FTH1 complex” section.

Answer 4: To address this issue, Fig. 1 has now been split into two figures; Fig. 1 focuses on the overall organization of the NCOA4^{FB}-FTH1 complex, and Fig. 2 provides a detailed view of the interactions stabilizing the complex. Accordingly, the panels of Fig. 1 now show: (a) Schematic representation of the domain organization of NCOA4, highlighting NCOA4^{FB} and the 16 aa we solved. (b) A high-resolution EM-map showing the position of NCOA4^{FB} relative to FTH1. (c) Panel (a) of the original Fig. 1, i.e., Schematic representation of the position of NCOA4^{FB} with respect to the geometry of the nanocage and the detailed molecular model of the interaction.

The panels of Fig. 2 now show: (a) Panel (b) of the original Fig. 1, i.e., the hydrophobicity of the NCOA4 binding cleft on FTH1. (b-c) A focused view of the hydrophobic pockets harboring the N and C-termini segment of NCOA4^{FB}, respectively.

Point 5:

Are FTH R23 and NCOA4 W497 engaged in a catio-pi interaction?

Answer 5: Preserving the backbone while selecting other rotamers for both R23 and W497 can lead to Pi interactions between these residues. However, the electron density of both residues, which are very clear in our map, suggests otherwise. Consequently, we conclude that in the stable conformation, the guanidine group of R23 is pointing away from the Pi system of the indole group of W497. It seems that these two amino acids are engaged by the hydrophobic effect between the long hydrophobic tail of R23 and the W497, while the guanidine group of R23 is interacting through a hydrogen bond with the carbonyl oxygen of S492, as described in this section.

Point 6:

Current writing might mislead the readers to believe the entire NCOA4 protein was solved in the cryoEM structure. The authors should make it clear in the revision where appropriate (especially in the abstract) that a NCOA4 small fragment was used and only a short 16 aa peptide was resolved. It might be useful to include a supplementary figure to depict the major domain/regions of the NCOA4 protein, fragment used in this study, and the resolved 16aa region. In this figure, the author could also consider including an AlphaFold model to show the intrinsically disordered nature.

Answer 6: To avoid a potentially misleading representation of our results, we:

- a. Changed the abstract according to the reviewers' request (as designated by Boldface letters) and it now reads:

The interaction between nuclear receptor coactivator 4 (NCOA4) and the iron storage protein ferritin is a crucial component of cellular iron homeostasis. The binding of NCOA4 to the FTH1 subunits of ferritin initiates ferritinophagy—a ferritin-specific autophagic pathway leading to the release of iron stored inside ferritin. The dysregulation of NCOA4 is associated with several diseases, including neurodegenerative disorders and cancer, highlighting the NCOA4-ferritin interface as a prime target for drug development. Here, we present the cryo-EM structure of the NCOA4-FTH1 interface, **resolving 16 amino acids of NCOA4 that are crucial for the interaction**. The characterization of FTH1 mutants, designed to modulate the NCOA4–FTH1 interaction, was used to validate the significance of the different features of the NCOA4 binding site. Our results explain the role of the large solvent-exposed hydrophobic patch found on the surface of FTH1 and pave the way for the rational development of ferritinophagy modulators.

- b. Added a panel to figure 1 that summarizes the published NCOA4 secondary structure predictions, on which we mapped the segment of 16 aa presented in this research. This is also stated in the figure legends:

Fig. 1a Schematic representation of the domain organization of NCOA4. The NCOA4^{FB} segment is in an orange box, and the 16 amino acids mediating the interaction with FTH1 are displayed.

Point 7:

Typo “dimmed”. Should be “deemed”?

“Relion” should be changed to RELION

Answer 7: We thank the reviewer for their meticulous reading—the spelling error and the naming of RELION were corrected.

Reviewer 2

We thank the reviewer for the supportive review of our manuscript and their useful comments. We agree with most of their points. Detailed responses and a list of changes we introduced to the text, in accordance with the review, are listed below each of the reviewer's comments.

1. The authors use the term "gorge" to describe the region of ferritin that interacts with NCOA4. This reviewer was confused by this term as it has not been used to describe ferritin or other protein structures. Other terms that might be clearer could be "cleft," "furrow" or "groove".

We accept the reviewer's point. We switched the term **gorge** to the more commonly used term **cleft**.

2. Would an image of the FTH-NCOA4 structure rotated so as to position NCOA4 on top of the FtnH be useful? This could be added to Figure 1.

We accept the reviewer's point. A similar point was also raised by reviewer 1. To address this issue, we split Fig. 1 into two figures. The overall position of NCOA4^{FB} on FTH1 is shown as an electron density map in Fig. 1b, and in highly detailed views in Fig. 2c-b.

3. The authors should put results in the main text and not in the Figure legends. The legend for Fig. 2 reads like results. Limit the legend to simply describing the possible position and interactions of the side chains and make clear these are models only, not structures.

The figure legends of Figure 2

We accept the reviewer's point. The legends are now describing the planned role of each mutation. We also made it clear that the panels showing the mutated amino acids are presumed rather than experimentally determined structures.

Accordingly, the following changes to the legends were introduced. Of note, the original Fig. 2 is now Fig. 3.

Original:

Figure 2: Visualization of the mutations designed for modulating the interaction of FTH1 with NCOA4 and the structure-guided design of the binding site for NCOA4 on FTL. (a-c) NCOA4 is orange, FTH1 is gray and mutated amino acids are in red, thick frames designate the *WT*. (a) A19Y: abolishes the FTH1-NCOA4 interaction by blocking the opening of the hydrophobic gorge that harbors NCOA4. The bulky Y residue is expected to clash with several amino acids of NCOA4 (aa494-498). (b) N22A: Removes one of the three H-bonds stabilizing the NCOA4-FTH1 interface. Because this H-bond is the only one protected from the solvent, it is expected to be the strongest of the three. Therefore, this mutation is expected to dramatically reduce the interaction. (c) R23A is known to abolish the interaction¹⁰. The structure explains this result, by the loss of an H-bond with S492 and of hydrophobic interaction with W497. R23I and R23L are expected to partially mitigate the strong effect of R23A by preserving part of the hydrophobic pocket found in the *WT* but abolishing the H-bond. (d) the hydrophobic surface of *WT*-FTL on the left, *WT*-FTH1 on the right, and the designed L2H mutant in the middle. The mutations are designated on each panel. Visualization of the phosphomimetic mutations S114D and S114E and a visualization of the designed changes to the electrostatic surface of the gain of function mutant L2H are depicted in Supplementary Figures 2 and 3 respectively.

Now reads:

Figure 3: Visualization of the mutations designed for modulating the interaction of FTH1 with NCOA4 and the structure-guided design of the binding site for NCOA4 on FTL. (a-c) NCOA4 is orange, FTH1 is gray and mutated amino acids are in red, **dashed frames designate the presumed position of the amino acids after mutations**. (a) A19Y: abolishes the FTH1-NCOA4 interaction by blocking the opening of the hydrophobic cleft that harbors NCOA4. (b) N22A: Removes one of the three H-bonds stabilizing the NCOA4-FTH1 interface (c) R23A: is known to abolish the interaction¹⁰. R23I and R23L: preserves part of the hydrophobic pocket found in the *WT* but abolishes the H-bond. (d) the hydrophobic surface of *WT*-FTL on the left, *WT*-FTH1 on the right, and the designed L2H mutant in the middle. The mutations are designated on each panel. Visualization of the phosphomimetic mutations S114D and S114E and a visualization of the designed changes to the electrostatic surface of the gain of function mutant L2H are depicted in Supplementary Figures S2 and S3 respectively.

4. Negative data are very important for the interpretation of the Ftn mutants. Include a table that reports the ITC data for all of the mutants in the main text of the paper, not just in the supplement. Can you also report dissociation constants? Cell biologists are more familiar with this kinetic measurement.

We thank the reviewer for this point. A table summarizing all results of binding assays including K_D values calculated from the ITC data was moved to the main text.

5. The authors report that the phosphomimetic mutants of FtnH S114 are only partially active for binding. Yet the data in Fig. 4 indicate that these mutants are not different from wild type FtnH in their binding activity and therefore suggest phosphorylation does not modulate this interaction and would not warrant further study. Why are the authors suggesting otherwise?

This point was also raised by Reviewer 1. The main reason for this unclear point in the text is the difference between the ITC measurements and the cell based binding assays, and the lack of a clear table that compiles all the binding results in a way that a reader can compare them. We thoroughly addressed this point by adding the Table as mentioned in point 4 (above) and by changes to the text as elaborated by the Answer 1 to Reviewer 1.

6. Text in the ITC images in supplement S4 are too small to read. Annoying.

The panels depicting the raw ITC data in Supplementary Fig. 4 are now large and clear.

7. Co-localization experiments are very good.

Thank you!

8. Discussion makes clear that this structure does not explain the 1:3 stoichiometry of Ftn-NCOA4 binding. The authors suggest that this is likely due to steric hindrance from disordered regions of the NCOA4 protomer. Can you model this at all?

We thank the reviewer for this point. We tried using various structure prediction tools. All of them predict that the region we are studying is unstructured. This is now shown in panel a of Fig. 1. Consequently, modeling a structure would be misleading in this case.

REVIEWERS' COMMENTS

Reviewer #1 (Remarks to the Author):

The revised manuscript has satisfactorily addressed my previous comments.

Reviewer #2 (Remarks to the Author):

The revisions in this excellent manuscript have addressed all my concerns